# UNIREL-R1: RL-TUNED LLM REASONING FOR KNOWLEDGE GRAPH RELATIONAL QUESTION ANSWERING

## ABSTRACT

Knowledge Graph Question Answering (KGQA) has traditionally focused on *entity-centric* queries that return a single answer entity. However, real-world queries are often relational, seeking to understand how entities are associated. In this work, we introduce *relation-centric* KGQA, a complementary setting where the answer is a subgraph capturing the semantic connections among entities rather than an individual entity. The main challenge lies in the abundance of candidate subgraphs, where trivial or overly common connections often obscure the identification of unique and informative answers. To tackle this, we propose *UniRel-R1*, a unified framework that integrates subgraph selection, multi-stage graph pruning, and an LLM fine-tuned with reinforcement learning. The reward function is designed to encourage compact and specific subgraphs with more informative relations and lower-degree intermediate entities. Extensive experiments show that UniRel-R1 achieves significant gains in connectivity and reward over Vanilla baselines and generalizes effectively to unseen entities and relations.

## 1 INTRODUCTION

Traditional knowledge graph question answering (KGQA) is largely *entity-centric*, aiming to return a single target entity. For example, a query such as "Who is Meghan Markle's husband's grandmother?" yields "Queen Elizabeth II" as the answer (see Figure 1(a)). Alternatively, users also often pose *relation-centric* queries that seek to uncover the relationships between entities rather than retrieve an individual fact. For example, "How are Meghan Markle and Queen Elizabeth II associated?" requires constructing a subgraph that captures their semantic connections. Such queries are common in real-world settings but remain beyond the scope of traditional KGQA systems.

Existing work in KGQA has primarily adhered to the entity-centric paradigm, often augmenting LLMs with structured knowledge to improve accuracy. Retrieval-augmented generation (RAG) incorporates KG facts as contextual input (Linders & Tomczak, 2025), while other approaches combine LLMs with graph neural networks (GNNs) for joint reasoning over text and structure (Xu et al., 2025; Yasunaga et al., 2021; He et al., 2024). To enhance interpretability, some methods generate explicit reasoning paths for multi-hop queries (Zhou et al., 2018; Chakraborty, 2024; Zhang et al., 2018), and others employ reinforcement learning (RL) or search strategies such as Monte Carlo Tree Search (MCTS) to explore candidate paths (Shen et al., 2025). Despite these advances, current systems ultimately return a single entity, leaving relation-centric queries unexplored.

In this work, we introduce *relation-centric* KGQA, a complementary setting to the standard *entity-centric* KGQA. Instead of producing an entity, the answer is a subgraph that captures the underlying relational structure among the seed entities. As demonstrated in Figure 1(a), multiple subgraphs may serve as answers. While Answer A1 is the shortest, it reflects an overly common relation and conveys little information, whereas Answers A2 and A3 provide more unique and informative connections. This illustrates the central challenge of relation-centric KGQA: Among many candidate subgraphs, trivial or generic ones often obscure unique and informative answers.

To address this challenge, we propose *UniRel-R1*, a unified framework for relation-centric KGQA. UniRel-R1 integrates *subgraph selection*, *multi-stage graph pruning*, and *RL-tuned LLM*. Figure 1(b) illustrates the pipeline, which begins by extracting seed entities from a natural language

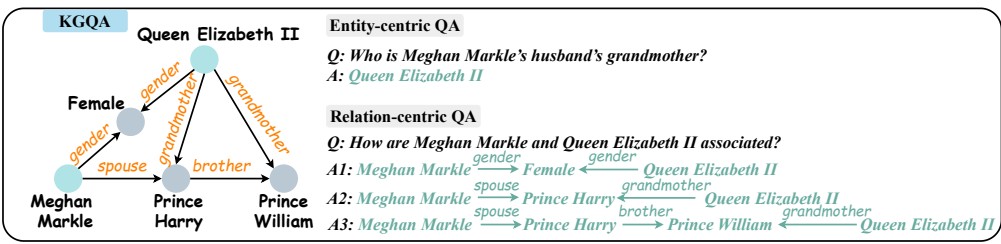

**(a) KGQA Paradigms: Entity-centric vs. Relation-centric**

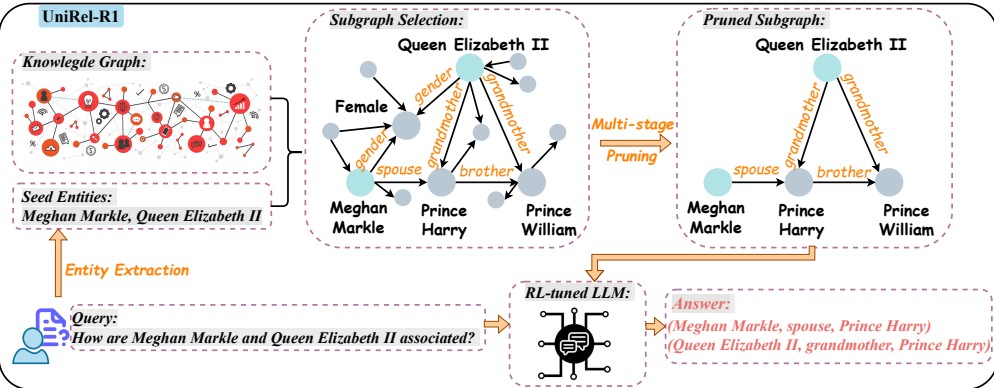

**(b) Pipeline of UniRel-R1**

Figure 1: An overview of KGQA and our proposed UniRel-R1 framework. (a) An illustrative example contrasting entity-centric KGQA with relation-centric KGQA. (b) The UniRel-R1 pipeline, which integrates subgraph selection, graph pruning, and RL-tuned LLM.

query. It then performs subgraph selection to identify candidate relational structures. A pruning stage follows to remove trivial or overly generic entities and relations, yielding a compact subgraph. The subgraph is textualized and provided alongside the query to an RL-tuned LLM, where the reward design encourages concise and informative outputs by favoring smaller structures with rarer relations and lower-degree intermediates. In this way, UniRel-R1 ensures that the final answers are not only valid but also capture distinctive and meaningful relations.

We conduct extensive experiments across seven benchmark knowledge graphs and a diverse set of LLMs from the Qwen and Llama families. To this end, we build relation-centric query sets covering both two-entity and multi-entity settings, evaluated under in-domain and cross-domain scenarios. Results show that *UniRel-R1* consistently outperforms Vanilla baselines, yielding at least a **35%** improvement in connectivity and over a **245%** gain in reward, while generalizing to unseen entities and relations. Interestingly, Qwen models are substantially more sensitive to the removal of semantic information than Llama models, highlighting a fundamental difference in how these families leverage semantic cues.

## 2 PROBLEM STATEMENT

In relation-centric KGQA, a knowledge graph is formally defined as $\mathcal{G} = (\mathcal{E}, \mathcal{R}, \mathcal{T})$, where $\mathcal{E}$ is the set of entities, $\mathcal{R}$ the set of relation types, and $\mathcal{T} \subseteq \mathcal{E} \times \mathcal{R} \times \mathcal{E}$ the set of triples. Given a query $q$, let $\mathcal{E}_q \subseteq \mathcal{E}$ denote the set of seed entities extracted from $q$. The task is to construct a subgraph:

$$\mathcal{G}^* = (\mathcal{E}^*, \mathcal{R}^*, \mathcal{T}^*) \subseteq \mathcal{G}.$$

A subgraph $\mathcal{G}^*$ is considered *valid* if it connects all entities in $\mathcal{E}_q$, with $\mathcal{E}_q \subseteq \mathcal{E}^*$, thereby ensuring that the answer is grounded in the semantics of $\mathcal{G}$. When multiple valid subgraphs exist, user preferences may vary depending on their background knowledge or task requirements. In this work, we prioritize subgraphs that are more *informative*, rather than those dominated by trivial or overly common structures. *Accordingly, our optimization objective balances compactness, relation rarity, and lower-degree intermediate entities*, ensuring that the resulting subgraphs emphasize non-trivial and distinctive semantic relations. Alternative optimization objectives can also be defined, depending on the specific application scenario.

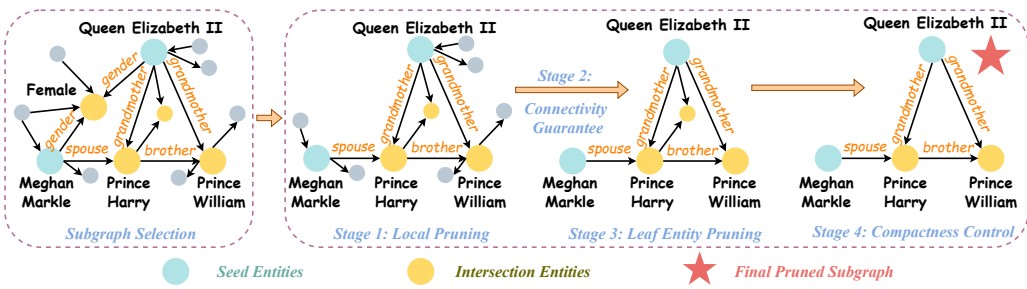

Figure 2: An example of multi-step graph pruning. From the initial subgraph (left), Stage 1 removes overly common entities (e.g., *Female*), Stage 2 enforces connectivity through intersections, Stage 3 prunes leaf entities, and Stage 4 enhances compactness by removing high-penalty intersection nodes (yellow). The final subgraph (right) retains the most informative relations among the seed entities.

## 3 FRAMEWORK: UNIREL-R1

In this section, we outline the overall pipeline of *UniRel-R1*, which consists of three stages. A task-specific subgraph is first constructed from the knowledge graph based on entities mentioned in the query (Sec. 3.1). The subgraph is then refined through multi-stage pruning to remove redundant or trivial entities and relations (Sec. 3.2). Finally, an RL-tuned LLM processes the textualized subgraph with the query to generate an answer that highlights the most informative relations(Sec. 3.3).[1]

### 3.1 SUBGRAPH SELECTION

The first stage of our framework constructs an initial candidate subgraph from the knowledge graph. Given a query $q$, the seed entity set is extracted from $q$ as $\mathcal{E}_q = \{e_1, e_2, \ldots, e_m\}$. For each seed entity $e_i \in \mathcal{E}_q$, we define its $k$-hop expansion as

$$\mathcal{V}^k(e_i) = \{v \in \mathcal{E} \mid \text{dist}(e_i, v) \leq k\}, \quad \mathcal{T}^k(e_i) = \{(u, r, v) \in \mathcal{T} \mid u, v \in \mathcal{V}^k(e_i)\},$$

where $\text{dist}(\cdot, \cdot)$ denotes the distance in the undirected projection of $\mathcal{G}$.

The candidate subgraph is obtained by aggregating the $k$-hop neighborhoods of all seed entities:

$$\mathcal{G}' = (\mathcal{V}', \mathcal{R}', \mathcal{T}'), \quad \mathcal{V}' = \bigcup_{e_i \in \mathcal{E}_q} \mathcal{V}^k(e_i), \quad \mathcal{T}' = \bigcup_{e_i \in \mathcal{E}_q} \mathcal{T}^k(e_i), \quad \mathcal{R}' = \{r \in \mathcal{R} \mid (u, r, v) \in \mathcal{T}'\}.$$

The subgraph $\mathcal{G}'$ serves as the initial candidate, refined through pruning in the next stage.

### 3.2 MULTI-STEP GRAPH PRUNING

The candidate subgraph $\mathcal{G}' = (\mathcal{V}', \mathcal{R}', \mathcal{T}')$ often contains redundant structures and overly generic entities that obscure meaningful relational patterns. To mitigate this issue, we refine $\mathcal{G}'$ through a principled multi-stage pruning process.

As a measure of entity generality, we introduce the *hub penalty*, defined for each $e \in \mathcal{E}$ as:

$$\text{HubPenalty}(e) = \log\big(1 + \deg(e)\big), \tag{1}$$

where $\deg(e)$ denotes the degree of $e$ in the undirected projection of $\mathcal{G}$. The penalty increases with entity degree, such that highly connected entities (e.g., *male*) receive larger values, while rarer and more specific entities (e.g., *president*) are penalized less. This formulation provides a principled criterion for filtering out uninformative entities and guiding the pruning of $\mathcal{G}'$.

Figure 2 shows the multi-stage pruning on the example in Figure 1, with each stage explained below.

**Stage 1: Local Pruning.** For each expansion set $\mathcal{V}^k(e_i)$, we impose a threshold $\rho$ on the hub penalty to eliminate overly common entities. Each **non-seed** $v$ with $\text{HubPenalty}(v) \geq \rho$ is removed along with its incident relations. Subsequently, entities that become isolated (i.e., with no incident relations) are discarded. The resulting pruned neighborhoods are denoted by $\mathcal{V}_\rho^k(e_i)$ and $\mathcal{T}_\rho^k(e_i)$.

---

[1]Note that the framework can be extended to alternative optimization objectives by modifying the pruning criteria and adjusting the reward function accordingly.

**Stage 2: Connectivity Guarantee.** Following local pruning, the filtered neighborhoods $\{\mathcal{V}_\rho^k(e_i)\}$ may fail to ensure connectivity among all seed entities. To evaluate connectivity, we compute pairwise intersections $\mathcal{I}_{ij} = \mathcal{V}_\rho^k(e_i) \cap \mathcal{V}_\rho^k(e_j)$ and construct an auxiliary graph $\mathcal{H}_\rho = (\{\mathcal{V}_\rho^k(e_i)\}, \{(i,j) \mid \mathcal{I}_{ij} \neq \emptyset\})$, where each node represents a pruned neighborhood $\mathcal{V}_\rho^k(e_i)$, and an edge $(i,j)$ is introduced whenever their intersection is non-empty. If $\mathcal{H}_\rho$ is connected, then all seed entities are jointly connected in the candidate subgraph. Otherwise, the threshold $\rho$ is incrementally relaxed and Stage 1 is reapplied until connectivity is restored.

**Stage 3: Leaf Entity Pruning.** Once the connectivity is ensured, we iteratively eliminate non-seed entities whose neighbor set $\mathcal{N}(v) = \{u \in \mathcal{V}' \mid \exists r \in \mathcal{R}' : (v,r,u) \in \mathcal{T}' \text{ or } (u,r,v) \in \mathcal{T}'\}$ contains only a single distinct element ($|\mathcal{N}(v)| = 1$). Such leaf-like entities and their incident relations are iteratively pruned until no further removal is possible.

**Stage 4: Compactness Control.** If the refined subgraph remains excessively large, we further enforce compactness by leveraging the intersections $\{\mathcal{I}_{ij}\}$ defined in Stage 2. Let $\mathcal{I}^+ = \{\mathcal{I}_{ij} \mid \mathcal{I}_{ij} \neq \emptyset\}$ denote the collection of all non-empty intersections. We then select a subset $\mathcal{I}^c \subseteq \mathcal{I}^+$ whose index pairs jointly cover all seed entities, i.e., $\bigcup_{\mathcal{I}_{ij} \in \mathcal{I}^c}\{i,j\} = \{0, 1, \ldots, m-1\}$. For each valid subset $\mathcal{I}^c$, a candidate subgraph is constructed in two steps. First, within each intersection $\mathcal{I}_{ij}$, we retain the $m$ entities with the lowest hub penalties:

$$\widehat{\mathcal{I}}_{ij} = \operatorname{argmin}_{\mathcal{U} \subseteq \mathcal{I}_{ij}, |\mathcal{U}|=m} \sum_{v \in \mathcal{U}} \mathrm{HubPenalty}(v).$$

Second, each selected set $\widehat{\mathcal{I}}_{ij}$ is expanded via a $k$-hop search restricted to the corresponding pruned neighborhoods: $\mathcal{N}^k(\widehat{\mathcal{I}}_{ij}) = \widehat{\mathcal{I}}_{ij} \cup \{u \in \mathcal{V}_\rho^k(e_i) \cup \mathcal{V}_\rho^k(e_j) \mid \mathrm{dist}(u,v) \leq k, \ v \in \widehat{\mathcal{I}}_{ij}\}$. The candidate node set associated with $\mathcal{I}^c$ is then $\mathcal{V}(\mathcal{I}^c) = \bigcup_{\mathcal{I}_{ij} \in \mathcal{I}^c} \mathcal{N}^k(\widehat{\mathcal{I}}_{ij})$, and its induced subgraph is subsequently simplified using Stage 3.

Finally, the most compact subgraph under parameters $(m, \rho)$ is obtained as:

$$\mathcal{G}_\rho^m = (\mathcal{V}_\rho^m, \mathcal{R}_\rho^m, \mathcal{T}_\rho^m),$$

where $\mathcal{V}_\rho^m = \operatorname{argmin}_{\mathcal{S} \in \mathcal{I}^+} |\mathcal{V}(\mathcal{S})|, \mathcal{T}_\rho^m = \{(u,r,v) \in \mathcal{T}' \mid u, v \in \mathcal{V}_\rho^m\}, \mathcal{R}_\rho^m = \{r \in \mathcal{R}' \mid (u,r,v) \in \mathcal{T}_\rho^m\}$.

**Stage 5: Iterative Reduction.** If the subgraph remains larger than the target size, the parameter $m$ is decreased, and Stage 4 is reapplied until a sufficiently compact subgraph is obtained.

### 3.3 Relational Answer Generation via LLM

Given the refined subgraph $\mathcal{G}_\rho^m$, the final step is to generate an answer that highlights the most informative relational structure. We employ an RL-tuned LLM to produce relational explanations conditioned on the query and textualized subgraph, with optimization based on RL algorithms and the task-specific reward function detailed in the following sections.

#### 3.3.1 Reinforcement Learning Algorithms

**Reinforcement Learning with Verifiable Reward (RLVR).** Our approach builds on the RLVR paradigm (Lambert et al., 2024), which applies to domains where the quality of generated responses can be deterministically verified. In its standard form, RLVR employs a rule-based verifier $v : \mathcal{X} \to \{0, 1\}$ that assigns binary rewards:

$$r_i = v(x_i) = \begin{cases} 1, & \text{if } x_i \text{ satisfies a task-specific correctness check,} \\ 0, & \text{otherwise.} \end{cases}$$

This binary scheme is effective for tasks with unambiguous success criteria (e.g., mathematical problem-solving and code generation). In our setting, we design a corresponding rule-based reward to verify model generated relations, described in Sec. 3.3.2.

**Group Relative Policy Optimization (GRPO).** For policy optimization, we adopt Group Relative Policy Optimization (GRPO) (Shao et al., 2024), which evaluates responses based on their relative performance within a sampled group from the policy.

Given a query $q$, we sample from the previous policy $\pi_{\theta_{\mathrm{old}}}(\cdot \mid q)$ to get $G$ responses $\{o_i\}_{i=1}^G$ with rewards $\{r_i\}$. The advantage $A_i$ is the group-normalized reward, defined as:

$$A_i = \frac{r_i - \overline{r}}{\sqrt{\frac{1}{G}\sum_{j=1}^{G}(r_j - \overline{r})^2 + \varepsilon_{\text{stab}}}}, \quad \overline{r} = \frac{1}{G}\sum_{j=1}^{G} r_j,$$

where $\varepsilon_{\text{stab}}$ ensures numerical stability. With $\rho_i = \frac{\pi_\theta(o_i|q)}{\pi_{\theta_{\text{old}}}(o_i|q)}$, the GRPO loss is defined as a clipped surrogate similar to PPO:

$$\mathcal{L}_{\text{GRPO}}(\theta) = -\left[ \frac{1}{G}\sum_{i=1}^{G} \min\big(\rho_i A_i, \text{clip}(\rho_i, 1-\alpha, 1+\alpha)\, A_i\big) - \beta\, D_{\text{KL}}\big(\pi_\theta(\cdot|q)\,\|\,\pi_{\text{ref}}(\cdot|q)\big) \right].$$

This objective encourages the policy $\pi_\theta$ to favor above-average responses. A $\beta$-scaled KL term regularizes the policy updates to prevent large deviations from a reference policy $\pi_{\text{ref}}$.

### 3.3.2 Reward design

In relation-centric KGQA, a valid answer must follow the required format, ensure connectivity, and emphasize informative entities and relations. To capture these aspects, we design a composite reward with four components: *Format*, *Connectivity*, *Entity Informativeness*, and *Relation Informativeness*.

Formally, for each answer $a$, we define:

**Format Reward.** The format reward is defined as $R_{\text{fmt}}(a) \in \{-1, 1\}$, where $R_{\text{fmt}}(a) = 1$ when $a$ conforms to the required format for parsing and $R_{\text{fmt}}(a) = -1$ otherwise.

**Connectivity Reward.** The connectivity reward evaluates whether the output subgraph connects the extracted entities $\mathcal{E}_q$ and is defined as $R_{\text{con}}(a) \in \{-\lfloor|\mathcal{E}_q|/2\rfloor, \ldots, \lceil|\mathcal{E}_q|/2\rceil - 1\}$, where the minimum indicates full disconnection, intermediate values reflect partial connectivity (e.g., $-\lfloor|\mathcal{E}_q|/2\rfloor + l$ for $(l+1)$-connected entities), and the maximum denotes full connectivity.

**Entity Informativeness Reward.** This component favors informative over generic entities. For a generated answer $a$ with entity set $\mathcal{E}(a)$, the reward is the sum of normalized hub penalties:

$$R_{\text{ent}}(a) = \sum_{e \in \mathcal{E}(a)} R_{\text{ent}}(e), \quad R_{\text{ent}}(e) = -\frac{\text{HubPenalty}(e)}{\max_{v \in \mathcal{E}} \text{HubPenalty}(v)} \in [-1, 0], \tag{2}$$

where $\text{HubPenalty}(e)$ is defined in Equation 1.

**Relation Informativeness Reward.** This component favors infrequent relations, which are more informative. The informativeness of a relation $r$ is measured by its inverse document frequency (IDF) (Robertson, 2004): $\text{IDF}(r) = \log\left(\frac{|\mathcal{T}|}{|\{(u,r,v) \in \mathcal{T}\}|}\right)$, where $|\mathcal{T}|$ is the number of triples in $\mathcal{G}$ and the denominator is the frequency of $r$. Frequent relations (e.g., *gender*) receive lower IDF scores, while rarer ones (e.g., *invention*) score higher.

Given an answer $a$ with relation set $\mathcal{R}(a)$, the reward is the sum of normalized IDF scores:

$$R_{\text{rel}}(a) = \sum_{r \in \mathcal{R}(a)} R_{\text{rel}}(r), \quad R_{\text{rel}}(r) = \frac{\text{IDF}(r)}{\max_{s \in \mathcal{R}} \text{IDF}(s)} - 1 \in [-1, 0]. \tag{3}$$

**Overall Reward.** The final reward for an answer $a$ aggregates all four components:

$$R(a) = R_{\text{fmt}}(a) + R_{\text{con}}(a) + \tfrac{1}{2}\left(\tfrac{R_{\text{ent}}(a)}{x} + \tfrac{R_{\text{rel}}(a)}{y}\right), \tag{4}$$

where $x$ and $y$ are normalization constants ensuring $\frac{R_{\text{ent}}(a)}{x}, \frac{R_{\text{rel}}(a)}{y} \in [-1, 0]$, with values determined by the maximum permitted subgraph size.

By construction, the reward is bounded as a function of the number of seed entities $|\mathcal{E}_q|$: $R(a) \in [-\lfloor|\mathcal{E}_q|/2\rfloor - 2, \lceil|\mathcal{E}_q|/2\rceil)$. If $R_{\text{fmt}}(a) = -1$, then $R(a) = -\lfloor|\mathcal{E}_q|/2\rfloor - 2$. If $R_{\text{con}}(a) = -\lfloor|\mathcal{E}_q|/2\rfloor$, then $R(a) = -\lfloor|\mathcal{E}_q|/2\rfloor$.

## 4 Experiment

In this section, we evaluate the effectiveness of *UniRel-R1* on relation-centric KGQA across multiple datasets and LLMs. We first describe the datasets and models, then present the main results, followed by analyses on generalization and on scalability through multi-entity queries.

### 4.1 EXPERIMENT SETUP

**Datasets.** We evaluate on seven benchmark KG datasets to ensure diversity in scale, domain, and relational complexity: Freebase13 (Socher et al., 2013), FB15k-237 (Toutanova & Chen, 2015), MetaQA (Zhang et al., 2018), DBpedia50/500 (Auer et al., 2007), YAGO3-10 (Mahdisoltani et al., 2015), and UMLS (Bodenreider, 2004). These datasets span encyclopedic, biomedical, and commonsense knowledge, providing a comprehensive testbed for relation-centric KGQA. Dataset statistics are reported in Appendix A.2.

**Query Construction.** Existing KGQA benchmarks focus on *entity-centric* queries that return a single entity. To adapt them to the relation-centric setting, we constructed 2,500 queries per dataset, with 2,000 for training and 500 for testing. For each dataset, we generated two-entity queries with $\text{dist}(e_i, e_j) \leq 4$ in the knowledge graph. To evaluate the scalability of *UniRel-R1*, we further created multi-entity queries (involving three and four entities) on DBpedia50. Full construction details are provided in Appendix A.2.

**Models.** To assess generalizability of our method, we evaluated *UniRel-R1* on LLMs from the Qwen (Yang et al., 2025) and Llama (Grattafiori et al., 2024) families, covering diverse architectures and scales. Specifically, we used Qwen-2.5-3B/7B/14B-Instruct, Llama-3.2-3B-Instruct, and Llama-3.1-8B-Instruct, spanning 3B–14B parameters. These results demonstrate the broad applicability and robustness of our approach across different model sizes and architectures.

**Prompts.** We designed structured prompts that encode the textualized knowledge graph as node and edge tables and incorporate the query, requiring the model to return the corresponding subgraph in a standardized triple format. This ensures parsable outputs that are directly comparable across models. Prompt templates are provided in Appendix A.2.

**Evaluation Metrics.** We evaluate performance using two primary metrics: *Connectivity Ratio* ($C$) and *Average Reward* ($\bar{R}$). The *Connectivity Ratio* measures the percentage of queries for which the generated subgraph successfully connects all seed entities, while the *Average Reward* reflects the overall subgraph quality defined in Equation 4. Additional metrics, such as individual reward components, are described in Appendix A.2.

### 4.2 MAIN RESULTS

**Parameter Choice.** For the normalization terms $x$ and $y$ in Equation 4, we calibrated their values using the Qwen-2.5-3B-Instruct model on the largest dataset, DBpedia500. To balance the contributions of the reward components, we set $x = 7$ and $y = 6$. These values were fixed and applied uniformly across all datasets and models to ensure consistency and comparability. Further details of this parameter tuning are provided in Appendix A.3.

Table 1 presents the performance of Vanilla and UniRel-R1 across datasets. Each entry is reported as $(C, \bar{R})$, where $C$ denotes the connectivity ratio and $\bar{R}$ the average reward. The last row shows the *Optimal Reward*, obtained via exhaustive search.

Overall, UniRel-R1 consistently surpasses Vanilla. Across datasets, it delivers at least a **35%** improvement in connectivity and a **245%** increase in average reward, underscoring its effectiveness in generating valid and informative subgraphs. Within both the Qwen and Llama families, larger models demonstrate stronger performance under Vanilla, reflecting the advantages of increased capacity. UniRel-R1 further amplifies these gains, highlighting the complementary benefits of model scaling and task adaptation.

In terms of model-specific results, Qwen-14B achieves the best outcomes on FB15k-237 and MetaQA, while Llama-8B performs best on the remaining datasets. On Freebase13 and DBpedia50, performance nearly matches the optimal reward, likely due to the relatively high connectivity of these graphs. Detailed reward breakdowns and additional metrics are provided in Appendix A.3.

**Case Study.** Beyond the quantitative results in Table 1, we present case studies on multiple datasets to qualitatively assess the outputs of *UniRel-R1*. Representative examples are shown in Figure 3, where our generated subgraphs are compared with those from the smallest subgraph baseline (i.e., with the fewest edges). The baseline often relies on trivial relations (e.g., *gender* or *place of birth*), whereas *UniRel-R1* highlights semantically richer intermediates (e.g., *influenced by Dante*

Table 1: Performance comparison between Vanilla and UniRel-R1 where each entry represents (connectivity ratio, average reward).

| Models | Datasets | Freebase13 | FB15k-237 | MetaQA | DBpedia50 | DBpedia500 | YAGO3-10 | UMLS |
|---|---|---|---|---|---|---|---|---|
| | | *Qwen Models* | | | | | | |
| Qwen3B | Vanilla | (41.2%, -0.53) | (0.6%, -1.41) | (0.4%, -1.30) | (11.2%, -1.10) | (0.6%, -1.43) | (1.0%, -1.4) | (23.0%, -0.73) |
| | UniRel-R1 | (86.0%, 0.64) | (62.0%, -0.02) | (72.8%, 0.07) | (91.0%, 0.55) | (64.8%, -0.06) | (67.4%, -0.03) | (69.0%, 0.08) |
| Qwen7B | Vanilla | (57.8%, 0.04) | (3.4%, -1.06) | (4.8%, -1.13) | (24.8%, -0.64) | (1.0%, -1.18) | (10.4%, -1.08) | (37.6%, -0.56) |
| | UniRel-R1 | (93.0%, 0.75) | (70.6%, 0.15) | (79.0%, 0.15) | (93.6%, 0.59) | (72.6%, 0.05) | (76.4%, 0.09) | (86.2%, 0.24) |
| Qwen14B | Vanilla | (72.0%, 0.24) | (17.8%, -0.88) | (11.6%, -0.87) | (37.8%, -0.55) | (6.8%, -1.18) | (16.6%, -0.84) | (41.8%, -0.41) |
| | UniRel-R1 | (97.2%, 0.83) | **(83.6%, 0.28)** | **(87.4%, 0.22)** | (98.4%, 0.67) | (82.8%, 0.14) | (89.8%, 0.22) | (97.6%, 0.36) |
| | | *Llama Models* | | | | | | |
| Llama3B | Vanilla | (0.0%, -2.97) | (0.0%, -2.97) | (0.0%, -2.92) | (0.0%, -3.00) | (0.0%, -2.92) | (0.0%, -2.93) | (0.0%, -2.92) |
| | UniRel-R1 | (95.8%, 0.80) | (70.6%, 0.14) | (70.6%, 0.12) | (60.6%, 0.04) | (38.8%, -0.47) | (75.4%, 0.09) | (34.6%, 0.36) |
| Llama8B | Vanilla | (1.0%, -2.89) | (0.4%, -2.74) | (0.0%, -2.80) | (0.0%, -2.96) | (0.0%, -2.67) | (0.2%, -2.51) | (0.0%, -2.87) |
| | UniRel-R1 | **(98.6%, 0.85)** | (82.6%, 0.27) | (82.6%, 0.20) | **(99.0%, 0.67)** | **(85.0%, 0.13)** | **(93.6%, 0.25)** | **(98.8%, 0.38)** |
| **Optimal Reward** | | 0.92 | 0.77 | 0.65 | 0.72 | 0.65 | 0.67 | 0.79 |

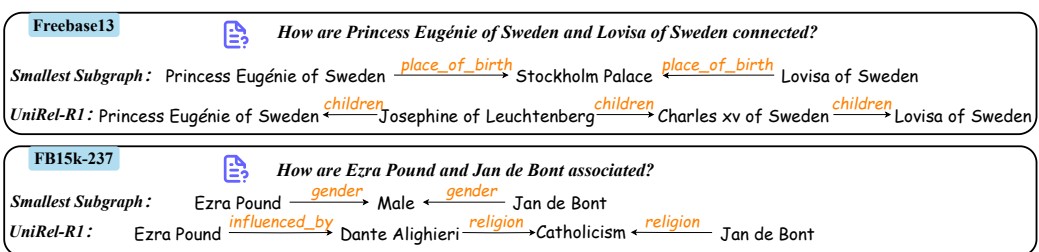

Figure 3: Case studies in Freebase13 and FB15k-237.

*Alighieri*) that yield more informative explanations. This improvement results from pruning, which filters generic nodes, together with RL optimization that promotes unique and meaningful structures. Additional case studies are provided in Appendix A.3.

### 4.3 GENERALIZATION

To further examine the robustness and transferability of *UniRel-R1*, we evaluate its ability to generalize across datasets. Table 2 reports the generalization results, where each tuple denotes $(C, \bar{R})$. Specifically, UniRel-R1 is trained on the largest dataset, DBpedia500, and then directly evaluated on the remaining datasets without additional training.

As shown in the *Original* rows, the models successfully generate valid subgraphs even for previously unseen entities and relations, demonstrating a notable degree of cross-domain transferability. Compared with the Vanilla results in Table 1, these models achieve substantially higher performance, in many cases approaching the connectivity and reward of their in-domain UniRel-R1 counterparts.

This observation motivates the hypothesis that LLMs can exploit semantic regularities acquired during training to generalize to novel patterns. To test this, we construct *Modified* versions of each dataset by replacing all entities and relations with random identifiers (e.g., $\text{ENT}1, \ldots, \text{ENT}n$, $\text{REL}1, \ldots, \text{REL}m$), thereby eliminating semantic cues.

As expected, the *Modified* rows exhibit a substantial drop in performance across all datasets for the Qwen family, confirming the central role of semantic information in enabling transferability. In contrast, the Llama family is less affected: The 3B model shows only marginal changes, and although the 8B model does experience reductions, the magnitude remains considerably smaller than that observed for Qwen models. For example, on YAGO3-10, Qwen-3B suffers reductions of **58.3%** in connectivity ratio and **266.67%** in average reward, whereas Llama-3B shows much smaller changes of 0.45% and 13.89%. Overall, these results indicate that Qwen models are more sensitive to the removal of semantic information than Llama models.

To further probe this effect, we conduct an additional experiment on MetaQA, where only a fraction of entities and relations are replaced with random identifiers. We consider partial modifications at 25%, 50%, and 75%, thereby progressively diminishing the amount of preserved semantic information. As shown in Figure 4, Qwen models exhibit a steady decline in both connectivity ratio and

Table 2: Cross-dataset generalization of UniRel-R1 trained on DBpedia500 where each entry represents (connectivity ratio, average reward).

| Datasets | Models | Qwen3B | Qwen7B | Qwen14B | Llama3B | Llama8B |
|---|---|---|---|---|---|---|
| Freebase13 | Original | (80.0%, 0.52) | (83.4%, 0.59) | (94.0%, 0.77) | (81.8%, 0.55) | (91.0%, 0.72) |
| | Modified | (69.6%, 0.33) | (74.8%, 0.43) | (84.0%, 0.60) | (81.6%, 0.55) | (87.0%, 0.65) |
| FB15k-237 | Original | (47.0%, -0.23) | (61.6%, 0.01) | (76.6%, 0.16) | (45.6%, -0.28) | (69.8%, 0.08) |
| | Modified | (20.8%, -0.71) | (41.4%, -0.32) | (52.4%, -0.12) | (45.8%, -0.25) | (59.4%, -0.03) |
| MetaQA | Original | (60.6%, -0.12) | (69.8%, 0.04) | (78.0%, 0.11) | (64.2%, -0.06) | (80.0%, 0.18) |
| | Modified | (29.2%, -0.56) | (49.0%, -0.21) | (60.8%, -0.02) | (67.2%, 0.06) | (72.6%, 0.15) |
| DBpedia50 | Original | (91.4%, 0.55) | (96.6%, 0.64) | (97.6%, 0.66) | (56.8%, -0.02) | (98.2%, 0.66) |
| | Modified | (54.6%, -0.06) | (73.0%, 0.27) | (84.0%, 0.45) | (84.6%, 0.45) | (91.8%, 0.57) |
| YAGO3-10 | Original | (56.6%, -0.18) | (67.4%, 0.01) | (80.6%, 0.14) | (44.8%, -0.36) | (83.0%, 0.18) |
| | Modified | (23.6%, -0.66) | (41.4%, -0.34) | (48.2%, -0.24) | (44.6%, -0.31) | (62.2%, -0.03) |
| UMLS | Original | (49.0%, -0.19) | (55.8%, -0.07) | (74.0%, 0.10) | (56.4%, -0.12) | (66.6%, 0.05) |
| | Modified | (41.0%, -0.30) | (43.4%, -0.24) | (50.0%, -0.14) | (49.4%, -0.17) | (55.0%, -0.07) |

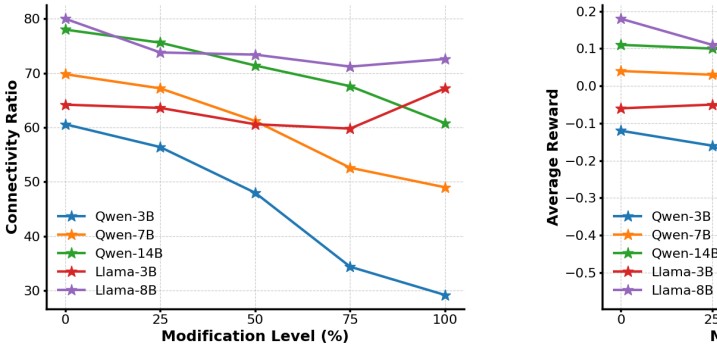

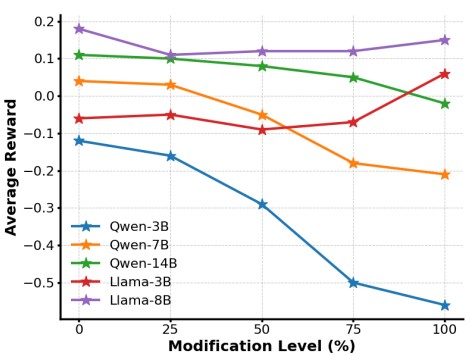

(a) Connectivity Ratio v.s. Modification Level      (b) Average Reward v.s. Modification Level

Figure 4: Comparison of connectivity ratio and average reward under different modification levels.

average reward as the modification level increases, indicating that their generalization performance is highly dependent on the semantic regularities of the knowledge graph. By contrast, Llama models are less affected: the 3B variant shows minor fluctuations, with a slight initial decline followed by a modest recovery, whereas the 8B model exhibits a mild downward trend with reductions of 9.25% in connectivity ratio and 16.67% in average reward between the original and fully modified settings.

Taken together, these findings suggest that Qwen models rely more on semantic information for cross-dataset generalization, whereas Llama models draw more on structural connectivity to sustain stable performance.

## 4.4 SCALABILITY

To assess the scalability of *UniRel-R1*, we extend our evaluation from two-entity queries to queries involving three and four entities on the DBpedia50 dataset. Unlike the two-entity case, where connectivity is measured by a *connectivity ratio*, multi-entity queries allow for a richer characterization.

For three-entity queries, we report both the *pairwise connectivity ratio* (the percentage of queries where *only* two entities are connected) and the *full connectivity ratio* (all three entities are connected). For four-entity queries, we distinguish three levels: (i) *pairwise connectivity ratio*, (ii) *triple connectivity ratio*, and (iii) *full connectivity ratio*.

Table 3 summarizes the performance of Vanilla and UniRel-R1 on DBpedia50 across three-entity and four-entity queries. Across both settings, Vanilla models exhibit limited ability to recover valid subgraphs. By contrast, UniRel-R1 demonstrates substantial generalization gains: for three-entity queries, the full connectivity ratio increases by 591.3% and the reward by 277.27%, while for four-entity queries, the corresponding gains reach 355.6% and 87.5%. Moreover, Llama models consistently outperform Qwen models, in some cases approaching the optimal reward, highlighting their stronger capacity to leverage structural information.

Table 3: Performance comparison of Vanilla and UniRel-R1 on DBpedia50 across three-entity and four-entity queries.

| Models | | Three Entities | | | Four Entities | | | |
|---|---|---|---|---|---|---|---|---|
| | | Full Conn. | Pairwise Conn. | Reward | Full Conn. | Triple Conn. | Pairwise Conn. | Reward |
| Qwen3B | Vanilla | 4.2% | 2.2% | -1.21 | 1.8% | 0.0% | 0.6% | -2.31 |
| | UniRel-R1 | 69.2% | 17.2% | 1.17 | 8.2% | 18.6% | 50.8% | -0.29 |
| Qwen7B | Vanilla | 10.4% | 6.6% | -0.66 | 5.2% | 2.0% | 4.8% | -0.84 |
| | UniRel-R1 | 73.4% | 18.0% | 1.27 | 55.4% | 16.4% | 18.0% | 0.78 |
| Qwen14B | Vanilla | 13.8% | 13.6% | -0.56 | 4.8% | 0.6% | 5.6% | -2.37 |
| | UniRel-R1 | 95.4% | 2.0% | 1.59 | 83.2% | 6.1% | 4.3% | 1.26 |
| Llama3B | Vanilla | 0.0% | 0.0% | -2.93 | 0.0% | 0.0% | 0.0% | -3.82 |
| | UniRel-R1 | 90.8% | 7.6% | 1.57 | 48.4% | 18.4% | 24.2% | 0.68 |
| Llama8B | Vanilla | 0.0% | 0.0% | -2.90 | 0.0% | 0.0% | 0.0% | -3.78 |
| | UniRel-R1 | 94.6% | 4.8% | 1.61 | 80.0% | 5.6% | 10.2% | 1.26 |
| **Optimal Reward** | | 1.74 | | | 1.63 | | | |

These findings collectively demonstrate the scalability of the proposed framework to more complex multi-entity scenarios and underscore its applicability to real-world KGQA tasks that involve reasoning over multiple entities.

## 5 RELATED WORK

**Knowledge Graph Question Answering.** Recent progress in KGQA has increasingly incorporated LLMs to exploit their strong natural language understanding. A prominent direction is to generate direct answers by enriching LLMs with KG evidence, most commonly via RAG frameworks that retrieve relevant triples as context (Linders & Tomczak, 2025). Another line of work integrates LLMs with GNNs to enable joint reasoning over text and structured knowledge (Xu et al., 2025; Yasunaga et al., 2021; He et al., 2024). In parallel, research emphasizing explainability focuses on explicit reasoning paths for multi-hop questions, where the answer requires traversing multiple triples (Zhou et al., 2018; Chakraborty, 2024; Zhang et al., 2018). Such methods often employ RL or search strategies, including MCTS, to iteratively refine candidate paths (Shen et al., 2025). Others adopt embedding-based reasoning approaches (Saxena et al., 2020; Shi et al., 2021), where multi-hop inference is captured implicitly through learned representations across multiple layers.

**Improving LLM Reasoning via RLVR.** Recent work has demonstrated RLVR's broad applicability across a wide spectrum of tasks, including mathematical and logical reasoning (Guo et al., 2025), code generation (Wang et al., 2025a), multi-modal reasoning (Huang et al., 2025b; Wang et al., 2025b; Li et al., 2025), structured data tasks like relation extraction (Dai et al., 2025b) and GUI navigation (Shi et al., 2025), and complex reasoning strategies such as parallel thinking (Zheng et al., 2025). Concurrent research focuses on improving the framework through two main directions: exploring training paradigms like self-play (Liu et al., 2025; Huang et al., 2025a) and test-time reinforcement learning (Zuo et al., 2025), and developing more effective RL algorithms such as DAPO (Yu et al., 2025), VAPO (Yue et al., 2025), and high-entropy guided optimization techniques (Dai et al., 2025a; Wang et al., 2025c; Zhou et al., 2025).

## 6 CONCLUSION

In this paper, we introduced relation-centric KGQA, where the goal is to return a subgraph that captures semantic associations among entities, a complementary setting to the standard entity-centric KGQA. We presented *UniRel-R1*, a unified framework that integrates subgraph selection, multi-stage pruning, and RL-tuned LLMs, where the reward design explicitly favors compact subgraphs with rarer relations and lower-degree intermediates, thereby steering the model toward concise and distinctive relational answers. Through extensive evaluations on seven benchmark knowledge graphs, we demonstrated that UniRel-R1 delivers substantial improvements over Vanilla baselines, achieving large gains in both connectivity and reward while maintaining strong generalization to unseen entities and relations. Our analysis further revealed a model-level distinction: Qwen models exhibit a stronger reliance on semantic cues, whereas Llama models maintain more stable performance by leveraging structural connectivity. Finally, our results on multi-entity queries confirm the scalability of UniRel-R1, showing that the framework extends naturally to complex reasoning tasks.

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

REPRODUCIBILITY STATEMENT

We ensure reproducibility of our work as follows. The prompts used in all experiments are provided in Appendix A.2, and a complete list of hyperparameters is included in Appendix A.1. All datasets and models employed in our experiments are publicly available. In addition, detailed descriptions of dataset modifications and query constructions are documented in Appendix A.2.

THE USE OF LARGE LANGUAGE MODELS

This manuscript was refined with the assistance of LLMs, which were used to improve clarity, grammar, and overall readability. The use of these models was restricted to language editing and did not influence the scientific content.

## A    DETAILS OF EXPERIMENTS

### A.1    HYPERPARAMETERS

We summarize the key hyperparameters used in our experiments below.

**Data / Input Configuration.**

- Max Prompt Length: 8192
- Max Response Length: 2048
- Validation Batch Size: 1024
- Seed: 1

**Optimization**

- Global Batch Size (Actor/Critic): 64
- Learning Rate: $1 \times 10^{-6}$
- Weight Decay: $1 \times 10^{-2}$
- Optimizer: AdamW (bf16)
- Max Gradient Norm: 1.0

**Rollout Configuration**

- Number of Rollouts: 5 (train), 1 (validation)
- Rollout Temperature: 1.0 (train), 0.6 (validation)
- Rollout Top-$p$: 1.0 (train), 0.95 (validation)
- Max Batched Tokens: 10240

**KL Control.** KL Coefficient ($\lambda_{KL}$): $1 \times 10^{-2}$

**Training Setup.** Total Epochs: 20 (2-entities), 40 (3-entities, 4-entities)

### A.2    IMPLEMENTATION DETAILS

**Datasets**    Table 4 summarizes the datasets used in our experiments, including their sources, number of entities, and number of triples. For FB15k-237, we replace machine identifiers with their corresponding human-readable entity names. To address the issue of long and complex relation names in Freebase, we transform each relation into a simplified alias with the assistance of ChatGPT, followed by manual verification. For example, the original relation `/film/actor/film./film/performance/film` is shortened to `acted_in`.

**Query Construction**    For *two-entity* queries, we first sampled pairs of seed entities such that the shortest path distance $\text{dist}(e_i, e_j) \leq 4$ in the knowledge graph. The sampling procedure (Algorithm 1) performs repeated random selection of a source entity and then selects a target entity reachable within $k$ hops. This ensures that queries follow the graph's local connectivity.

To evaluate scalability, we further constructed queries involving *three* and *four* entities on the DB-pedia50 dataset. For triplets $(u, v, w)$, we required that the third entity $w$ is within $k$ hops of at least

Table 4: Statistics of knowledge graph datasets used in our experiments.

| Datasets | Source | #Entities | #Relations | #Triples |
|---|---|---|---|---|
| Freebase13 | Freebase | 75,043 | 13 | 375,514 |
| FB15k-237 | Freebase | 14,265 | 237 | 310,116 |
| MetaQA | Movie-domain | 40,151 | 9 | 134,741 |
| DBpedia50 | Subset of DBpedia (Wikipedia) | 30,449 | 365 | 43,756 |
| DBpedia500 | Larger subset of DBpedia | 490,598 | 573 | 4,268,614 |
| YAGO3-10 | Subset of YAGO3 (Wikipedia, WordNet, GeoNames) | 123,182 | 37 | 1,089,040 |
| UMLS | Unified Medical Language System | 135 | 46 | 6,529 |

---

**Algorithm 1** Entity Pair Sampling

---

1: Randomly sample an entity $u$ from the graph.
2: Run BFS up to $k$ hops to obtain reachable set $\mathcal{N}(u, k)$.
3: Randomly sample $v \in \mathcal{N}(u, k)$ with $d(u, v) \leq k$.
4: Return $(u, v)$ as the entity pair.

---

one of the first two entities ($u$ *or* $v$). For quadruplets $(u, v, w, z)$, we extended this requirement so that the fourth entity $z$ is within $k$ hops of at least one of $\{u, v, w\}$. This OR-connectivity guarantees that multi-entity queries yield semantically well-connected subgraphs.

After sampling seed entities, we converted them into natural language questions using a set of manually designed templates. For example, given the pair (*Meghan Markle*, *Queen Elizabeth II*), we generate the query: *"How are Meghan Markle and Queen Elizabeth II associated?"* We also used alternative phrasings such as *"For what reason are [entity1] and [entity2] related?"* to improve linguistic diversity.

**Prompts** For each experiment, we constructed prompts that contextualize the knowledge graph and the query into a unified text input. Specifically, the graph is represented as **node and edge tables**, followed by the natural language query, together with task-specific instructions. This design ensures that the model has access to both structural and semantic information.

An example prompt corresponding to Figure 2 is shown below.

```
You are given a directed graph as two CSV-like sections in this order:

1) Node table (header included):
node_id, node_attr

2) Edge table (header included):
src, edge_attr, dst

Task
- Use ONLY edges from the Edge table to answer the question by outputting
  a path.
- When printing each edge, replace IDs with the exact node_attr from the
  Node table.
- Output MUST be text triples, not numeric IDs.

Output format (STRICT | no extra text):
PATH:
("subject"|predicate|"object")
...
END

Rules
- Use only listed edges; do NOT invent edges.
- Map IDs → node_attr; preserve node_attr exactly.
- Output NOTHING outside the PATH block.
- If no path exists, output exactly:
PATH:
```

```
END

Graph:
node_id, node_attr
1, Meghan Markle
2, Prince Harry
3, Prince William
4, Queen Elizabeth II

src, edge_attr, dst
1, spouse, 2
4, grandmother, 2
4, grandmother, 3
2, brother, 3

Question: How are Meghan Markle and Queen Elizabeth II associated?

Your output must be ONLY the PATH block.
```

**Additional Evaluation Metrics**   Besides the two primary metrics, Connectivity Ratio and Average Reward, we report additional reward components to provide a more fine-grained evaluation of relational answers:

- *Correct Format Percentage* ($F$): the proportion of generated subgraphs that conform to the required output format.
- *Average Entity Informativeness* ($I_E$): the *negative* of the average informativeness score of entities (Equation 2); lower values indicate more informative entities.
- *Average Relation Informativeness* ($I_R$): the *negative* of the average informativeness score of relations (Equation 3); lower values indicate more informative realtions.

Note that *Average Entity Informativeness* and *Average Relation Informativeness* are only available when the format reward equals 1 and the generated subgraph is connected. Therefore, for two-entity queries, we compute these averages only over the connected answers.

A.3   DETAILS OF MAIN RESULTS

Table 5: Evaluation results of Qwen-2.5-3B-Instruct model on DBpedia500 under different parameter settings. Only reachable cases are reported for $I_R$ and $I_E$.

| $x$ | $y$ | $F$ | $C$ | $I_R$ | $I_E$ |
|---|---|---|---|---|---|
| 5 | 4 | 100.0% | 59.0% | -5.12 | -5.84 |
| 6 | 5 | 100.0% | 60.6% | -5.62 | -6.43 |
| 7 | 6 | 100.0% | 64.8% | -5.83 | -7.24 |
| 8 | 7 | 100.0% | 64.2% | -6.39 | -7.95 |

**Parameter Choice.**   For the normalization terms $y$ and $x$ in Equation 4, we calibrated their values using the Qwen-2.5-3B-Instruct model on the largest dataset, DBpedia500. Since the shortest distance between entity pairs is bounded by $\text{dist}(e_i, e_j) \leq 4$, the optimal case for two-entity queries typically involves about four relations and five entities. Accordingly, we initialized the parameter search from $(x, y) = (5, 4)$.

Table 5 presents the evaluation across different $(x, y)$ values. Transitioning from $(5, 4)$ to $(6, 5)$ leads to a marginal improvement in Connectivity Ratio of 1.6%, but this gain is offset by a reduction in informativeness, with $I_R$ decreasing by 0.50 and $I_E$ by 0.59, resulting in a total loss of 1.09. By comparison, $(7, 6)$ achieves the highest Connectivity Ratio of 64.8%, representing a 4.2% increase over $(6, 5)$, while the informativeness loss remains essentially identical: $I_R$ decreases by 0.21 and $I_E$ by 0.81, again summing to 1.09. This outcome highlights that $(7, 6)$ delivers a substantially better trade-off, offering a larger connectivity gain without incurring additional informativeness penalties.

Table 6: Average relation and entity informativeness of Vanilla and UniRel-R1 where each tuple represents $(I_R, I_E)$.

| Models | Datasets | Freebase13 | FB15k-237 | MetaQA | DBpedia50 | DBpedia500 | YAGO3-10 | UMLS |
|---|---|---|---|---|---|---|---|---|
| | | | | *Qwen Models* | | | | |
| Qwen3B | Vanilla | (0.59, 0.67) | (1.93, 1.93) | (1.19, 1.12) | (0.91, 0.53) | (1.31, 1.225) | (0.60, 0.26) | (1.08, 1.61) |
| | UniRel-R1 | (0.34, 0.94) | (2.97, 2.28) | (2.90, 2.45) | (2.01, 1.78) | (3.109, 4.04) | (2.78, 4.69) | (1.25, 4.78) |
| Qwen7B | Vanilla | (0.32, 0.75) | (1.63, 1.79) | (2.52, 1.63) | (1.07, 0.71) | (2.14, 1.519) | (1.88, 1.43) | (1.10, 2.06) |
| | UniRel-R1 | (0.36, 1.19) | (2.01, 2.84) | (2.90, 2.71) | (2.02, 1.79) | (3.12, 4.20) | (2.81, 4.81) | (1.34, 6.26) |
| Qwen14B | Vanilla | (0.33, 0.87) | (2.06, 2.52) | (1.93, 0.50) | (1.46, 1.05) | (3.00, 3.00) | (1.82, 1.84) | (1.38, 2.58) |
| | UniRel-R1 | (0.35, 1.20) | (1.69, 4.18) | (2.78, 3.81) | (1.99, 1.98) | (2.99, 5.19) | (2.46, 5.68) | (1.42, 6.71) |
| | | | | *Llama Models* | | | | |
| Llama3B | Vanilla | (-, -) | (-, -) | (-, -) | (-, -) | (-, -) | (-, -) | (-, -) |
| | UniRel-R1 | (0.35, 1.23) | (2.14, 2.87) | (2.14, 2.87) | (1.89, 1.69) | (3.05, 4.22) | (2.89, 4.57) | (0.62, 1.58) |
| Llama8B | Vanilla | (0.07, 3.07) | (3.87, 6.82) | (-, -) | (-, -) | (-, -) | (4.17, 11.62) | (-, -) |
| | UniRel-R1 | (0.36, 1.29) | (1.97, 4.23) | (1.97, 4.23) | (2.02, 1.96) | (3.03, 5.78) | (2.78, 6.08) | (1.43, 6.83) |
| **Optimal Search** | | (0.29, 0.77) | (1.12, 1.86) | (2.66, 1.78) | (2.08, 1.45) | (2.58, 1.92) | (2.526, 1.67) | (0.91, 1.93) |

Table 7: Correct format percentage of Vanilla and UniRel-R1.

| Models | Datasets | Freebase13 | FB15k-237 | MetaQA | DBpedia50 | DBpedia500 | YAGO3-10 | UMLS |
|---|---|---|---|---|---|---|---|---|
| | | | | *Qwen Models* | | | | |
| Qwen3B | Vanilla | 83.6% | 78.8% | 84.6% | 84.6% | 78.2% | 78.0% | 93.0% |
| | UniRel-R1 | 100.0% | 99.8% | 95.4% | 100.0% | 99.8% | 100.0% | 100.0% |
| Qwen7B | Vanilla | 96.6% | 93.8% | 89.4% | 95.0% | 90.0% | 86.8% | 88.8% |
| | UniRel-R1 | 100.0% | 100.0% | 95.8% | 100.0% | 100.0% | 100.0% | 100.0% |
| Qwen14B | Vanilla | 93.4% | 91.2% | 94.0% | 88.2% | 85.4% | 94.2% | 93.8% |
| | UniRel-R1 | 100.0% | 100.0% | 95.6% | 100.0% | 100.0% | 100.0% | 100.0% |
| | | | | *Llama Models* | | | | |
| Llama3B | Vanilla | 1.6% | 1.6% | 4.2% | 0.4% | 3.8% | 3.4% | 3.8% |
| | UniRel-R1 | 100.0% | 99.8% | 99.8% | 100.0% | 100.0% | 100.0% | 100.0% |
| Llama8B | Vanilla | 4.4% | 12.6% | 10.2% | 2.0% | 16.6% | 24.6% | 4.6% |
| | UniRel-R1 | 100.0% | 100.0% | 100.0% | 100.0% | 100.0% | 100.0% | 100.0% |

When further increasing the parameters to $(8, 7)$, the Connectivity Ratio ceases to improve and instead begins to decline.

Based on these observations, we selected $(y, x) = (7, 6)$ as the final configuration. These values were fixed and applied uniformly across all datasets and models with two-entity queries to ensure consistency and comparability.

**Performance Comparison across Models and Datasets.** Table 6 reports the average relation and entity informativeness of Vanilla and UniRel-R1 across datasets, where lower values indicate more informative relations or entities.

For Vanilla, the connected answers are typically simple subgraphs with shorter lengths, which explains the relatively low initial scores. After applying UniRel-R1, additional relations and entities may be introduced to ensure connectivity, which can increase the informativeness scores compared to the optimal baseline.

Compared with the optimal search, certain datasets (e.g., Llama-8B on DBpedia50) exhibit lower $I_R$ but higher $I_E$, revealing a gap between relation- and entity-level informativeness. By contrast, datasets with larger reward gaps from the optimal (e.g., FB15k-237) often show a small difference in $I_R$ but a large difference in $I_E$, suggesting that extra high-degree or hub-penalty entities were included to achieve connectivity.

Table 7 further reveals that Qwen models exhibit a stronger ability to follow prompt instructions under Vanilla, maintaining relatively high correct format percentages (often above 80%). In contrast, Llama models struggle to produce valid outputs in the absence of structural guidance, with correct format rates frequently below 5%. With UniRel-R1, however, both families achieve nearly perfect compliance across all datasets, indicating that reinforcement-guided adaptation is highly effective in enforcing structural answers.

**Case Study.** We present three additional case studies to further illustrate the differences between the shortest subgraph and the UniRel-R1.

**Question:** How are *Football* and *University of Rome La Sapienza* related?

**Shortest Subgraph:**

$$\text{Football} \xrightarrow{\text{has\_country}} \text{Italy} \xrightarrow{\text{contains}} \text{University of Rome La Sapienza}$$

**UniRel-R1:**

$$\text{Football} \xleftarrow{\text{team\_sport}} \text{A.S. Roma} \xrightarrow{\text{team\_location}} \text{Rome} \xleftarrow{\text{headquarter\_city}} \text{University of Rome La Sapienza}$$

**Question:** How are *Dianne Wiest* and *Villanova University* related?

**Shortest Subgraph:**

$$\text{Dianne Wiest} \xrightarrow{\text{nationality}} \text{United States of America} \xrightarrow{\text{contains}} \text{Villanova University}$$

**UniRel-R1:**

$$\text{Dianne Wiest} \xrightarrow{\text{student}} \text{University of Maryland, College Park} \xrightarrow{\text{fraternities\_and\_sororities}} \text{Alpha Delta Pi}$$
$$\xrightarrow{\text{fraternities\_and\_sororities}} \text{Villanova University}$$

**Question:** How are *2003 NFL Draft* and *Philip Baker Hall* related?

**Shortest Subgraph:**

$$\text{2003 NFL Draft} \xrightarrow{\text{school}} \text{University of Miami} \xrightarrow{\text{contains}} \text{United States of America} \xleftarrow{\text{nationality}} \text{Philip Baker Hall}$$

**UniRel-R1:**

$$\text{2003 NFL Draft} \xrightarrow{\text{school}} \text{Marshall University} \xrightarrow{\text{fraternities\_and\_sororities}} \text{Alpha Sigma Phi}$$
$$\xrightarrow{\text{fraternities\_and\_sororities}} \text{University of Toledo} \xrightarrow{\text{student}} \text{Philip Baker Hall}$$