# OpenReview forum: "UniRel-R1: RL-tuned LLM Reasoning for Knowledge Graph Relational Question Answering"
_ICLR.cc/2026/Conference — ICLR 2026 Conference Withdrawn Submission_

### Official Review · Reviewer_vPor · 2025-10-22

**Soundness:** 2
**Presentation:** 2
**Contribution:** 2
**Rating:** 2
**Confidence:** 3

**Summary:**

The paper introduces UniRel-R1 for ``relation-centric'' KGQA: build a k-hop subgraph around seed entities, apply multi-stage pruning (hub-penalty thresholding, connectivity recovery by relaxing ρ, leaf pruning, and a compactness step with parameter m), then use reinforcement learning to finetune an LLM with a composite, rule-verifiable reward. Results are reported on seven KGs with gains over “Vanilla”, prompting in connectivity ratio, and the authors’ Average Reward metric.

**Strengths:**

- The paper points out the existing limitations of entity-centric KGQA and argues for subgraph-centric answers, which I think is very valuable.

- The subgraph punning methods are straightforward to use, and the reward definitions are valid and well-designed.

- Using a verifiable reward to steer LLM outputs toward structural validity is a good direction.

**Weaknesses:**

**Major**:

- **Benchmarking**: The main concern is that the evaluation relies on constructed relation-centric query sets and a self-defined reward as the main quality metric. There are no objective metrics, such as accuracy or F1, to measure the performance.  This risks reward hacking and undermines comparability as a benchmark.

- **RL necessity not demonstrated.** Because pruning plus (m, ρ) strongly constrains the candidate, the RL-tuned LLM operates in a narrow space defined by hyperparameters. This means that the operation space for RL is quite limited and does not affect the subgraph extraction and selection process, which is the main challenge.

- **Limited Baseline**: “Vanilla” LLM prompting is not competitive. Vanilla LLMs do not know the reward criterion, which puts them at a natural disadvantage during the evaluation that is solely based on reward comparison, thus giving UniRel an unfair advantage. Also missing comparisons to classic path-based or graph-optimization baselines (e.g., shortest/Steiner trees with rarity penalties, or RL agents over KGs) undercuts claims of superiority.

- **Application on traditional entity-centric KGQA**: It would be better to show the generalizability of UniRel if the author conducts additional experiments on traditional entity-centric KGQA.

- **Stage-2 trial-and-error and cost.** The Connectivity Guarantee relaxes \rho iteratively and re-runs Stage-1 until connected which is potentially expensive; no complexity analysis or early-stopping/scheduling is given.

**Minor**:
- **Typo**: line 276. Testbed => Tested
- **“Informativeness” is under-specified at selection time.** HubPenalty/IDF drive pruning/reward, but “more informative” is not formalized as a selection objective (vs. being enforced post-hoc by thresholds and reward).
- **Mismatch between problem and output (path-only)**. Although the task is defined as returning a subgraph, the prompt forces a single PATH block (“Use ONLY edges … by outputting a path … PATH: ... END), excluding trees/unions/cycles and narrowing expressivity.
- **Lack of released codebase.** The paper does not provide any implementation or repository link, making it difficult to assess the reproducibility of the proposed pipeline, query construction, and RL training setup.

**Questions:**

- Can you provide any other objective metrics to quantify the gain from the RL training?
- How do your selection methods work on traditional entity-centric KGQA, where metrics and baselines are available?
- What is the complexity/runtime of the Stage-2 ρ-relaxation loop? Can you provide a scalability analysis?
- Add prompted “Vanilla+reward-instruction” baselines (LLM explicitly told the reward criteria) to separate the effect of knowing the metric from learning it.

---

### Official Review · Reviewer_eNhY · 2025-10-30

**Soundness:** 3
**Presentation:** 3
**Contribution:** 3
**Rating:** 6
**Confidence:** 5

**Summary:**

This paper presents relation-centric KGQA, where the answer to a natural-language query is a subgraph connecting seed entities rather than a single entity. They have defined a three stage framework for relation-centric KGQA, it integrates subgraph selection, multi-stage graph pruning, and RL-tuned LLM. It begins by extracting seed entities from a natural language query. It then performs subgraph selection to identify candidate relational structures. A pruning stage follows to remove trivial or overly generic entities and relations, yielding a compact subgraph. The subgraph is textualized and provided alongside the query to an RL-tuned LLM, where the reward design encourages concise and informative outputs by favoring smaller structures with rarer relations and lower-degree intermediates. In this way, UniRel-R1 ensures that the final answers are not only valid but also capture distinctive and meaningful relations.

**Strengths:**

This paper presents the idea of shifting Knowledge Graph Question Answering (KGQA) from entity centric to relation centric. The framework (UniRel-R1) is well-structured, it integrates subgraph selection, multi-stage graph pruning, and RL-tuned LLM reasoning in a coherent pipeline.

**Weaknesses:**

The evaluation could be strengthened by including comparisons with stronger or more diverse baselines. Theoretical analysis or insights for the algorithm on its converges toward informative subgraphs will enhance its depth.

**Questions:**

What is the performance of UNIREL-R1 in comparison to the other baseline.

---

### Official Review · Reviewer_JfkY · 2025-10-31

**Soundness:** 2
**Presentation:** 3
**Contribution:** 2
**Rating:** 2
**Confidence:** 3

**Summary:**

This paper investigates "relation-centric KGQA," an under-explored direction in Knowledge Graph Question Answering (KGQA). Traditional KGQA focuses on "entity-centric" queries (e.g., "Who is X?") aiming to return a single entity. In contrast, the "relation-centric" queries defined in this work (e.g., "How are X and Y associated?") aim to return a **subgraph** that explains the semantic connections between entities. The core challenge lies in the vast number of candidate subgraphs, where most connections are trivial or overly broad (e.g., associated via "gender: female"). To address this, the authors propose **UniRel-R1**, a unified three-stage framework. The framework first performs subgraph selection (k-hop expansion), followed by a sophisticated **multi-stage graph pruning** algorithm to filter noise, which introduces a "Hub Penalty" to remove high-frequency (low-information) intermediate nodes. Finally, an **LLM tuned with Reinforcement Learning (RL)** processes this pruned, textualized subgraph to generate the final answer. The RL reward function is carefully designed to prefer subgraphs that are more compact and contain rarer relations and lower-degree intermediate nodes. Experiments on seven benchmark knowledge graphs show that UniRel-R1 significantly outperforms "Vanilla" baselines in connectivity and reward (e.g., >245% reward increase) and generalizes effectively to unseen entities and relations.

**Strengths:**

1. **Novel and Practical Problem Formulation and Framework:** The paper clearly defines and addresses the "relation-centric KGQA" problem. This is not only an important complement to traditional entity-centric KGQA but also aligns closely with real-world user needs to explore complex "why" and "how" queries. The UniRel-R1 framework designed for this task is a logically clear, end-to-end three-stage pipeline (select-prune-generate) that cleverly combines classic graph algorithms (heuristic pruning) with the reasoning capabilities of modern LLMs (RL tuning), providing a strong paradigm for solving this problem.

2. **Sophisticated and Effective Multi-Stage Pruning Mechanism:** A core contribution of this paper is its multi-stage graph pruning algorithm (Sec 3.2). By introducing a "Hub Penalty" (based on node degree) and "Relation Informativeness" (based on IDF) in the reward function, the framework effectively distinguishes "information-rich" connections from "trivial" ones.

3. **Significant Empirical Results and Strong Generalization:** The method consistently and substantially (e.g., >245% reward increase) outperforms the "Vanilla" baseline (Table 1) across seven benchmark knowledge graphs of varying domains and scales. This demonstrates the method's effectiveness and robustness.

**Weaknesses:**

**1. Insufficient Baselines and Unclear Comparative Setup:** The study's most significant weakness is its comparison against a "Vanilla" LLM baseline, which appears to be a weak "strawman." The paper (Table 1) reports massive gains over this baseline. However, the exact setup for this "Vanilla" baseline is ambiguous; the paper does not explicitly state whether it also benefited from the crucial Stage 2 pruning. If it operated on the unpruned k-hop subgraph, the comparison is fundamentally unfair. A convincing evaluation *must* include two missing key baselines: first, a crucial ablation baseline, `Vanilla LLM + Pruned Subgraph (Stage 2 output)`, which is necessary to validate the *true* additive contribution of the RL tuning (Stage 3); and second, SOTA comparison baselines, as the paper lacks comparisons to non-LLM graph algorithms (e.g., k-shortest paths or Steiner tree approximations) and other strong LLM-KG methods (e.g., `Think-on-Graph` [1], `KG-CoT` [2]).

**2. Conflated Contributions: Graph Pruning vs. Reinforcement Learning:** Closely related to the previous point, the paper's title and abstract frame the "RL-tuned LLM" as a core contribution. However, the framework's massive performance gains (i.e., improving from near-zero connectivity to ~80-90%) likely stem significantly from the Stage 2 'multi-step graph pruning' algorithm. This sophisticated pruning process (Stage 2) provides a highly-refined candidate set to the LLM, significantly narrowing the search space and embedding strong structural priors. This suggests the RL task may be substantially simplified by the pruning stage. The paper fails to provide the necessary ablation studies to disentangle the contributions of these two stages—for example, by systematically analyzing the independent impact of each pruning stage or each component of the reward function—casting doubt on the validity of its central claims.

**3. Framework Fragility and Sensitivity to Heuristics:** UniRel-R1 is a complex, multi-stage pipeline whose performance relies heavily on a series of hand-crafted heuristics and 'meta-hyperparameters'. For instance, the pruning stage (Stage 2) involves multiple parameters like $k$ for the k-hop, $\rho$ for the Hub Penalty threshold, and $m$ for Compactness Control. The reward stage (Stage 3) uses composite reward weights $x$ and $y$ (Table 5) that were tuned on only one model and one dataset in a very limited search. This makes the framework highly 'brittle' because these heuristic rules (especially the Hub Penalty threshold and reward normalization) are non-transferable. The significant variance in degree distribution and relation frequency across different KGs means that expensive re-tuning may be required every time the graph is changed.

**4. Questionable Scalability and Evaluation Realism:** The paper avoids several real-world challenges. First, regarding graph scalability, the paper's claim (Sec 4.4) is based only on tests on DBpedia50 (~30k nodes), ignoring the 'k-hop explosion' problem: on industrial-scale KGs (billions of nodes), a $k=4$ expansion is computationally infeasible. Second, query realism is questionable; the test set (Appx A.2) was built by pre-sampling entity pairs guaranteed to be $\le 4$ hops apart, creating an artificially 'easy' evaluation environment far removed from open-domain queries where path lengths are unknown or non-existent. Third, the paper does not adequately discuss its high inference latency, as every query requires a slow and expensive pipeline of retrieval, pruning, textualization, and generation. Finally, the framework systematically fails on incomplete KGs: the `R_con` (connectivity reward) penalizes outputs that are factually correct but where the graph is missing an edge—a fatal flaw for real-world KGs.

**5. Task Definition and Metric Misalignment:** The paper defines "relation-centric KGQA" as a new paradigm, but it highly overlaps with 'explainable KGQA' or 'multi-hop reasoning'. More critically, its core reward function is misaligned with semantic correctness. The reward function prioritizes structural attributes (correct format, connectivity, low-degree nodes, rare relations) over semantic utility. This leads to potential 'Reward Hacking': the model may be incentivized to select a "rare but marginal" path to "game" the score, while systematically penalizing 'common but crucial' paths (e.g., `place_of_birth`), thereby sacrificing the answer's true informativeness and explanatory power.


**References:**
> Sun, Jiashuo, et al. "Think-on-Graph: Deep and Responsible Reasoning of Large Language Model on Knowledge Graph." ICLR 2024
>
> Zhao, Ruilin, et al. "Kg-cot: Chain-of-thought prompting of large language models over knowledge graphs for knowledge-aware question answering." IJCAI 2024.

**Questions:**

None

---

### Official Review · Reviewer_3sqo · 2025-11-03

**Soundness:** 2
**Presentation:** 3
**Contribution:** 1
**Rating:** 6
**Confidence:** 3

**Summary:**

The paper points out that most KGQA work is entity-centric—it takes a question and returns one target node—but real users often ask relation-centric questions like “How are X and Y connected?” which require returning a subgraph that captures meaningful relationships, not just one entity. The authors formalize this relation-centric KGQA setting and note a key difficulty: KGs contain many trivial or overly generic connection paths (e.g., through very high-degree nodes), so naïvely returning the shortest path often gives uninformative answers. They propose UniRel-R1, a unified pipeline that (1) extracts seed entities from the query, (2) selects candidate subgraphs, (3) prunes trivial/high-degree/generic nodes and relations, and (4) feeds a textualized version to an RL-tuned LLM whose reward favors compact, distinctive, and rare-but-relevant subgraphs. On seven KGs and multiple LLM families (Qwen, Llama), UniRel-R1 improves connectivity and reward scores, and shows different sensitivity patterns across models.

**Strengths:**

1.Clearly identifies and formalizes a realistic but under-served setting, returning an informative subgraph instead of a single entity. It makes the contribution complementary to classic KGQA.

2.The multi-stage selection + pruning + RL reward that prefers rarer relations is well aligned with the core problem (too many generic connections).

3.Evaluating on seven KGs, two- vs multi-entity queries, and two LLM families strengthens the generalization claim; the observation about Qwen vs Llama sensitivity is a nice, nontrivial finding.

**Weaknesses:**

1.the proposed subgraph selection pipeline is general and very important for any KG reasoning or QA tasks. I suggest to conduct more studies on more benchmarks, to test its effectiveness.

2.Important baselines are not discussed and compared, e.g., KG-Agent and Plan-on-graph

**Questions:**

Please refer to the weakness part.

**Details Of Ethics Concerns:**

Please refer to the weakness part

---

### Author Response · Authors · 2025-11-18

Thank you very much for all the reviewers’ valuable comments. We will incorporate them into our future revisions. After careful consideration, we have decided to withdraw the submission.

---

### Note · Authors · 2025-11-18

I have read and agree with the venue's withdrawal policy on behalf of myself and my co-authors.